# Dual Plasticizer/Thermal Stabilizer Effect of Epoxidized Chia Seed Oil (*Salvia hispanica* L.) to Improve Ductility and Thermal Properties of Poly(Lactic Acid)

**DOI:** 10.3390/polym13081283

**Published:** 2021-04-14

**Authors:** Ivan Dominguez-Candela, Jose Miguel Ferri, Salvador Cayetano Cardona, Jaime Lora, Vicent Fombuena

**Affiliations:** 1Instituto de Seguridad Industrial, Radiofísica y Medioambiental (ISIRYM) Universitat Politècnica de València (UPV), Plaza Ferrándiz y Carbonell s/n, 03801 Alcoy, Spain; ivdocan@doctor.upv.es (I.D.-C.); scardona@iqn.upv.es (S.C.C.); jlora@iqn.upv.es (J.L.); 2Technological Institute of Materials (ITM), Universitat Politècnica de València (UPV), Plaza Ferrándiz y Carbonell 1, 03801 Alcoy, Spain; joferaz@upvnet.upv.es

**Keywords:** PLA, epoxidized chia seed oil (ECO), plasticizers, migration, disintegration

## Abstract

The use of a new bio-based plasticizer derived from epoxidized chia seed oil (ECO) was applied in a poly(lactic acid) (PLA) matrix. ECO was used due to its high epoxy content (6.7%), which led to an improved chemical interaction with PLA. Melt extrusion was used to plasticize PLA with different ECO content in the 0–10 wt.% range. Mechanical, morphological, and thermal characterization was carried out to evaluate the effect of ECO percentage. Besides, disintegration and migration tests were studied to assess the future application in packaging industry. Ductile properties improve by 700% in elongation at break with 10 wt.% ECO content. Field emission scanning electron microscopy (FESEM) showed a phase separation with ECO content equal or higher than 7.5 wt.%. Thermal stabilization was improved 14 °C as ECO content increased. All plasticized PLA was disintegrated under composting conditions, not observing a delay up to 5 wt.% ECO. Migration tests pointed out a very low migration, less than 0.11 wt.%, which is to interest to the packaging industry.

## 1. Introduction

Currently, a global plastics production of 368 million tons was recorded in 2019, an increment of 2.5% from 2018. Conventional polymers such a polyethylene (PE), polypropylene (PP), polyvinyl chloride (PVC), poly(ethylene terephthalate) (PET), polystyrene (PS), and polyamide (PA), represent approximately 70% of plastic demand in Europe. With regard to the packing industry, which represents around 40% of total demand, these polymers are leading the plastic demand [1]. The majority are non-biodegradables as well as manufactured by petrochemical industries (non-renewable resources) [2]. The food packaging industry generates a large volume of waste due to it short-lifespan and its recycling is often limited to those not contaminated with food products. According to Plastics Europe 2020, about 39.5% of post-consumer waste was recycled, while 18.5% (3.2 million tons) ended up in landfills [1]. These non-recycled plastics need to be managed to avoid the presence in seas, lakes, and rivers which threatens the environment [3,4].

Concerning biodegradable polymers, their presence is increasing in the food packing industry. Several biodegradable polymers such as poly(lactic acid) (PLA), thermoplastic polyurethane (TPU), and polyhydroxyalkanoates (PHAs) have been applied as new alternatives [5,6,7,8]. The most employed polymer is PLA (about 10.9%), which is obtained by fermentation of polysaccharides or sugar extracted from potato, sugarcane, corn, etc., thus obtained by renewable resources [9]. PLA is currently manufactured for common applications such as salad cups, lamination films, drinking cups, containers, etc. [10]. Biodegradation of PLA is produced by hydrolysis, resulting in harmless and non-toxic substances [11,12]. Nowadays, PLA is considered economically competitive and its properties such as good processability, high transparency, water solubility resistance, biodegradability, recyclability, etc., make it suitable for good packaging [13,14]. Besides, the energy saved in production is around 22–55% with regard to petroleum-based polymers, thus contributing to a decrease in environmental impact [15]. PLA is characterized by its high tensile modulus, although some drawbacks such as brittle nature with elongation at break of less than 9%, a narrow processing window, poor thermal stability, etc., are detected [16,17]. Several methodologies to improve ductile properties have been carried out successfully using copolymers, blends, or plasticizers in a PLA matrix [18,19,20].

The plasticizer market is increasing its annual demand in the polymeric industries. Phthalates (PTs) are a common plasticizer and additive to provide transparency, flexibility, and durability properties to a polymer matrix, commonly found in food packaging, medical equipment, building materials, toys, etc. [21,22]. The annual world production of PTs as plasticizer is approximately 80% [21]. However, studies show a migration phenomenon from polymer matrix to element in contact causing health and environment impact. It is well known that exposure of PTs to human lives produces endocrine damage, and reproduction and carcinogenic effects [23]. According to the European Union and other organizations, specific PTs such as diisobutyl phthalate (DIBP) or diethylhexyl phthalate (DEHP) among others, are banned for contents above 0.1 wt.% [24]. Among other alternatives, poly(ethylene glycol) (PEG), polyethylene oxide (PEO), and adipates are widely studied in a PLA matrix, obtaining an excellent improvement of ductile properties [25,26,27]. However, petrochemical-based plasticizers are questioned for toxicity and therefore there is a continuous attempt to obtain bio-based plasticizers [28].

Vegetable oils (VOs) are an interesting route to achieve renewable plasticizers for three main reasons: they are widely available, have a low toxicity, and are biodegradable. Two reactive sites are identified in fatty acids of vegetable oils to bring compatibility with a polymer matrix: double bonds and ester groups [29]. To increase this compatibility, vegetable oil can be epoxidized, which consists of introducing epoxy groups (oxirane ring) in double bonds. Several studies reported the use of epoxidized vegetable oils in PLA matrix, thus obtaining a bio-based polymer with high performance as a plasticizer. Some epoxidized oils such as epoxidized linseed oil (ELO) and epoxidized soybean oil (ESBO) are available commercially at competitive prices. Several studies using epoxidized oil with a non-elevated number of oxirane groups have been reported. The study performed by Qiong Xu et al. reported an improvement of elongation at break from 3.98 to 6.5% using 9 wt.% of ESBO [30]. Further percentage led to a decrease in ductile properties due to plasticizer saturation. Garcia-Garcia et al. found an increment of elongation at break from 7.8 to 15% with 5 wt.% of epoxidized Karanja oil [31]. In respect to impact energy, an evident improvement of 32% was obtained, confirming an effective plasticization. More interesting results were found by Balart et al., who reported an increment of 450% of elongation at break with respect to neat PLA employing ELO with 8% of epoxy groups [32].

Chia seed oil (CO) is a promising VO characterized by its high iodine value (above 190 g I_2_/100 g oil) [33], becoming suitable to be epoxidized. Epoxidized chia seed oil has not been previously applied in a polymer matrix and could present an elevated epoxy content, improving the compatibility between PLA.

As different authors have reported that the interaction between PLA and epoxidized chia seed oil (ECO) could occur between the carbonyl group, from ester linkage, present in the PLA main chain and the epoxy group of ECO. This reaction mechanism was proposed by Al-Mulla et al. [34]. Although the interaction mechanism is not very strong, the terminal location of the hydroxyl groups in PLA gives them a high availability to react with the epoxy groups [30]. Thus, based on these previous studies, Figure 1 shows the chemical interaction between PLA and ECO. This new bio-based plasticizer could be an alternative to ELO and ESBO plasticizers. For this reason, the aim of this work is studying the potential of epoxidized chia seed oil as a new bio-based plasticizer for PLA to be used in the packaging sector. Mechanical and thermal properties were tested to evaluate the effect of different percentage of ECO in a PLA matrix. A migration test was evaluated as an important property in the packaging sector. Finally, a disintegration test was carried out to evaluate the effect of ECO in polymer degradation.

## 2. Materials and Methods

### 2.1. Materials and Sample Preparation

Poly(lactic acid) (PLA) with a commercial grade 2003D was supplied by NatureWorks LLC (Minnetonka, MN, USA). Its density was 1.24 g·cm^−3^, with an approximate molecular weight of 120,000 g·mol^−1^. The base of the plasticizer was chia oil extracted through cold pressing extraction using a CZR-309 press machine (Changyouxin Trading Co., Zhucheng, China) from edible chia seed (*Salvia hispanica*, L.) supplied by Frutoseco (Bigastro, Alicante, Spain). The oil was characterized by a density of 0.928 g cm^−3^ at 25 °C and iodine value of 197. Epoxidation process was carried out in situ with hydrogen peroxide (30% *v/v*), acetic acid (99.7%), and sulfuric acid (97%) supplied by Sigma Aldrich (Sigma Aldrich, Madrid, Spain). Epoxidized chia seed oil (ECO) presented an epoxy content of 6.71%, equivalent to 238 EEW (Equivalent Epoxy Weight). This value was obtained following the guidelines of the ASTM D1652–97. ECO presented an iodine value of 25 and density of 1.026 g·cm^−3^ at 25 °C, which made it suitable to be used as plasticizer.

In the first stage, PLA pellets were dried at 60 °C for 24 h to remove moisture. After this, PLA pellets and ECO were weighed according to the compositions indicated in Table 1. Percentages of ECO were selected, taking into account published articles that consider that more than 10% of oil shows signs of plasticizer saturation [35,36,37]. All the five compositions selected were extruded at constant speed (40 rpm) using a twin-screw co-rotating extruder from DUPRA S.L (Castalla, Alicante, Spain) with the following temperature profiles: 162.5 (feeding zone), 165, 170, and 175 °C (die). After extrusion, samples were air-cooled to room temperature, pelletized, and dried again at 60 °C for 24 h for further processing by injection. Each composition was shaped by injected molding using a Meteor 270/75 from Mateu & Solé (Barcelona, Spain) at temperature profiles of 170, 180, 190, and 200 °C from feed section to injection nozzle. Cavity filling time was 1 s and the cooling time was set to 10 s.

### 2.2. Mechanical and Thermal Characterization

The effect of ECO on mechanical properties was studied using standard tensile, impact test, and hardness. Tensile properties were obtained in a universal test machine ELIB 30 from S.A.E Ibertest (Madrid, Spain). Five different samples were tested using 5 kN load cell and a crosshead speed of 10 mm·min^−1^ according to ISO 527. Furthermore, an axial extensometer from S.A.E Ibertest (Madrid, Spain) was used to obtain tensile modulus with high accuracy. The impact strength was tested in a 1 J Charpy pendulum from Metrotec S.A. (San Sebastián, Gipuzkoa, Spain), as is indicated in ISO 179. Shore D hardness was carried out following the guidelines of the ISO 868 using a durometer 673-D from J. Bot S.A (Barcelona, Spain). In both tests, a minimum of 5 samples were used and the results shown are the obtained average.

Thermomechanical properties were assessed by using standard Vicat softening temperature (VST) by using a load of 5 kg and a heating rate of 50 °C·h^−1^ according to ISO 306. Moreover, heat deflection temperature (HDT) was obtained following the guidelines of ISO 75 with a load of 296 g and a heating rate of 120 °C·h^−1^. Both values were obtained in a Vicat-HDT station mod. DEFLEX 687-A2 from Metrotec S.A. (San Sebastián, Gipuzkoa, Spain). At least five different specimens for each composition were tested and average values were calculated.

Additionally, storage modulus (G’) and damping factor (tan δ) were evaluated in torsion mode by dynamic mechanical thermal analysis (DMTA) in an oscillatory rheometer AR-G2 from TA instrument (New Castle, DE, USA). Shaped samples (4 × 10 × 40 mm^3^) were evaluated with a dynamic program from 30 to 110 °C using a heating rate of 2 °C·min^−1^. The maximum deformation was set to 0.1% with a constant frequency of 1 Hz.

Thermal transitions of PLA plasticized with different contents of ECO were obtained by differential scanning calorimetry (DSC) in a DSC mod. 821 from Mettler-Toledo Inc. (Schwerzenbach, Switzerland). Samples with an average weight of 6–8 mg were subjected to the following temperature program: 1st heating program from 25 to 300 °C at 10 °C·min^−1^ to remove thermal history, 2nd cooling program from 300 to 25 °C at 10 °C·min^−1^, and 3rd heating program from 25 to 300 °C at 10 °C·min^−1^. All thermal cycles were performed in a nitrogen atmosphere with a flow rate of 66 mL·min^−1^. The percentage of crystallinity of different PLA formulations with ECO was determined by Equation (1):(1)$$X_{c}\;\left(\%\right)=\left[\frac{∆H_{m}-∆H_{cc}}{∆H_{m\left(100\%\right)}}\right]\cdot \frac{1}{w_{sample}}\times 100$$
where $∆H_{cc}$ and $∆H_{m}$ represent cold crystallization and melting enthalpies (J·g^−1^), respectively. The weight amount of PLA is represented by $w_{sample}$ (g). Theoretical melting of PLA 100% crystalline ($∆H_{m\left(100\%\right)}$) was 93 J·g^−1^, as is reported [10].

Thermogravimetric analysis was carried out in a TGA/SDTA 851 thermobalance from Mettler-Toledo Inc (Schwerzenbach, Switzerland). A heating ramp from 30 to 700 °C at constant heating rate of 10 °C·min^−^^1^ and constant flow rate of nitrogen (66 mL·min^−^^1^) were set to evaluate samples with average weight of 10 mg. The temperature when a 5% weight loss has been reached and the maximum degradation were obtained in order to evaluate thermal stability of the different samples of PLA plasticized with ECO.

### 2.3. Morphology Characterization

Fractured surface from impact test was observed by field emission scanning electron microscopy (FESEM) model ZEISS ULTRA 55 from Oxford Instruments (Oxfordshire, UK). Previously fractured surfaces were coated with a thin metallic layer (Au-Pd alloy) employing a sputter coater EM MED020 from Leica Microsystems (Wetzlar, Germany) to avoid electrical charging. All samples were observed using an acceleration voltage of 2 kV.

### 2.4. Disintegration under Composting Conditions

Disintegration test was conducted in aerobic conditions according to ISO 20200 at temperature of 58 °C and a relative humidity of 55% using a synthetic compost reactor (300 × 200 × 100 mm^3^). Sample sizing of 25 × 25 × 1 mm^3^ were placed in a carrier bag and buried in controlled soil. Previously, all films manufactured with neat and plasticized PLA with ECO were dried at 40 °C over 24 h. Seven different samples of each formulation were employed for the disintegration process under composting conditions. Each control day (3, 7, 14, 17, 21, 24, and 28 days), a different sample of each formulation was unburied while the rest remained in the process. The removed samples were washed with distilled water and dried 24 h before to be weighed in an analytical balance. All tests were carried out in triplicate to ensure reliability. Average disintegration percentage of extracted samples was calculated using Equation (2).
(2)$$W_{l}\;\left(\%\right)=\frac{w_{0}-w}{w_{0}}\cdot 100$$
where $w_{0}$ is referred to initial dry weight of the sample and w is the weight of the sample extracted from compost soil on different days after drying. Furthermore, optical images were taken to record the progression of disintegration along time.

### 2.5. Migration of Plasticizer by Solvent Extraction Test

Migration test was studied by solvent extraction using n-hexane solvent as is indicated by several reports [36,38,39]. Samples of PLA and PLA plasticized with ECO were immersed in n-hexane solvent at different temperatures (30, 40, 50, and 60 °C) over 8 h in an air circulating oven mod. Selecta 2001245 by JP Selecta S.A. (Barcelona, Spain). Before and after experiments, all samples were dried at 40 °C for 24 h to ensure the absence of solvent. The weight loss of plasticizer ($WL_{p}$) obtained by migration test was calculated using Equation (3).
(3)$$WL_{p}\;\left(\%\right)=\frac{w_{b}-w_{a}}{w_{b}}\cdot 100$$
where $w_{b}$ is the weight of samples before experiment and $w_{a}$ is the weight after migration test.

## 3. Results and Discussion

### 3.1. Effect of ECO in PLA on the Mechanical Properties

One of the main disadvantages of PLA is its low ductile property, which gives it a characteristic brittleness. As is shown in Figure 2, neat PLA used in the present study reaches an elongation at break of 8% and a tensile strength higher than 45 MPa. The resulting tensile modulus was higher than 3100 MPa, as plotted in Figure 3. The plasticization effect was clearly observed in the different formulations of PLA with ECO. For example, the addition of 2.5 wt.% of ECO (PLA_2.5%ECO) provided higher ductile properties, with an elongation at break around 18%. Consequently, mechanical properties like tensile strength and tensile modulus slightly decreased (40.9 and 3040 MPa, respectively), which was attributed to the elastomeric and toughening effect of ECO plasticizer. In Figure 2, it was possible to observe how the slope of the stress–strain curve became lower as the ECO content in the samples increased. This decrease resulted in a lower tensile modulus. At the same time, the elongation at break increased notably, reaching values of 64.5% for samples with 10 wt. % of ECO. As a consequence of this variation in mechanical and ductile properties, the toughness of the samples was clearly improved. This increase represents a 700% rise compared to neat PLA. Therefore, the presence of epoxy groups in ECO interacts with hydroxyl groups present in PLA, decreasing intermolecular forces and, consequently, increasing its ductile properties [30]. These results are in accordance with previous studies on the use of epoxidized vegetable oils like PLA plasticizer. Yu-Qiong et al. reported an increase of 123% employing epoxidized palm oil [30]. This lower increase in elongation values was due to the lower content of epoxy groups in palm oil (3.23%) with respect to chia oil (6.71%), which is one of the main parameters to be taken into account. For example, other authors have reported similar increases of 700% or even higher using epoxidized vegetable oils with content of epoxy groups around 5.8% [36,40,41]. Thus, stronger interactions occurred as epoxy content increased due to the increased presence of reactive groups Therefore, in view of the results obtained, it seems that once epoxy group values of 5.8% or higher were reached, a saturation effect was shown, not reaching further increases in the elongation at break. Where the use of ECO with PLA appeared to have a direct effect was on the attenuating effect of the sharp drop in mechanical properties. For example, Chieng et al. reported that using palm oil in 10 wt.% proportion reduces the tensile strength of the neat PLA by almost 50%. However, this decrease was only 21% when ECO was employed [35]. Therefore, the higher oxirane oxygen content in ECO than, for example in palm oil, can interact more strongly with PLA chains leading to intense polymer-plasticizer interactions allowing to maintain mechanical and ductile balanced properties. As a consequence of these interactions between PLA and ECO, it was remarkable that samples with 7.5 and 10 wt.% ECO content provided constant values in tensile mechanical properties. On the other hand, this attenuation of decrease in mechanical properties also suggested, as other cited authors have pointed out, that a percentage higher than 10 wt.% produces a negative effect on ductile properties due to the excess of plasticizer and a possible phase separation. Some authors like Sanyang et al. and Rizzuto et al. remarked that once all free volume is full of plasticizer, a decrease of elongation at break occurs due to free volume reduction [42,43]. Similar results were reported by Emad et al., who observed a decrease of elongation at break above of 9 wt.% of epoxidized palm oil [44]. Therefore, PLA_10%ECO obtained the highest elongation at break (64.3%), together with a reduction of 6.7% in tensile modulus (2893 MPa) and 21% in tensile strength (33.4 MPa) in respect to initial value (PLA neat).

Other techniques that provide substantial information about the plasticizing effect of ECO on PLA are the Charpy impact test and Shore D hardness. As shown previously, the addition of ECO provides a plasticizing effect. As a result, tensile strength and tensile modulus decrease slightly with increment of ECO, contrary to the elongation at break. Impact absorbed energy is a parameter related to the ductile properties and toughness. Therefore, as expected initially, PLA is a brittle polymer, which has a low impact absorption capacity (37.1 J·m^−2^), but adding ECO improves absorbed impact energy due to elastomeric and toughing effect (Figure 4). It is possible to observe how an increase from 2.5 to 10 wt.% of ECO provides a gradual gain in impact absorption capacity. Sample ECO_10%ECO obtained the highest value (68.3 J·m^−2^), which represented an increase of 85% with respect to non-plasticized PLA. These results were in concordance with Carbonell-Verdu et al., who evaluated the plasticization effect on PLA using a 7.5 wt.% of epoxidized cottonseed oil, obtaining an increase on the impact of absorbed energy of 18% [45]. Again, the greater content of epoxy groups in ECO (6.7%) compared to cottonseed oil (5.8%), provided higher ductile properties. On the other hand, the plasticizing effect of ECO inversely affected the hardness of the samples. As the ECO content increased, the hardness decreased. However, in the same way that the tensile strength analysis showed a stabilization in the decrease of the properties of ECO_7.5%ECO and ECO_10%ECO samples, the difference between the hardness of neat PLA and ECO_10%ECO sample was only less than 6%.

Regarding the morphological changes produced by the incorporation of ECO to PLA, Figure 5a showed a brittle morphology with smooth surface and very low plastic deformation characteristic of neat PLA. With increasing ECO content, remarkable changes can be observed. Figure 5b, which represents PLA_2.5%ECO, showed a slight change in surface roughness. PLA_5%ECO, Figure 5c, shows a rougher surface as well as presence of filaments thus indicating an increase in ductility as a consequence of plasticization. So, an increase in roughness and a higher filament density was observed with increasing ECO. However, a higher presence of plasticizer (equal or more than 7.5 wt.%) began to display some spherical voids due to the plasticizer saturation, Figure 5d,e. A similar finding was reported by Ferri et al., who observed spherical voids above 5 wt.% of maleinized linseed oil as a plasticizer for PLA [46]. Phase separation was produced, and a worse miscibility occurred [31]. Finally, in Figure 5f, the presence of high density of filaments and voids in sample PLA_10%ECO, indicating a plasticizer saturation, can be observed better (2500×).

### 3.2. Effect of ECO in PLA on the Thermomechanical Properties of PLA

Storage modulus (G’) and damping factor (tan δ) were assessed by dynamic mechanical response. In Figure 6, the viscoelastic behavior of PLA_ECO formulations were exposed. Two characteristic changes in the storage modulus could be distinguished. The first change, between 50 and 70 °C, was the drop of storage modulus, which was related to glass transition temperature (T_g_) at around 60 °C, as reported by Yong et al. [47]. The second change, between 80 and 100 °C, was recognized as the beginning of cold crystallization process. The addition of ECO to PLA resulted in a loss of storage modulus at lower temperatures. This was due to the plasticizing effect that ECO exerts on the PLA matrix, which increases the free volume between PLA chains before a saturation effect, decreasing the interaction between them [48]. In addition, at room temperature, neat PLA showed a storage modulus value of 1300 MPa, while the plasticized PLA formulations showed a decrease of this modulus up to 1000 MPa as a consequence of the plasticization effect. On the other hand, the beginning of cold crystallization decreased as ECO content increased, obtaining a shift from 87 up to 84 °C for plasticized PLA. This effect was due to plasticizer enabling the rearrangement in packed structure under lower energetic conditions.

The temperature of the tan δ peak was a great manner to obtain an accurate value of T_g_ which was moved to lower temperatures as ECO content increased. Specifically, the T_g_ values were reduced from 64.2 (PLA neat) up to 61.9 and 59.5 °C for PLA_5%ECO and PLA_10%ECO, respectively. Regarding tan delta peak magnitude, it is related to their molecular mobility. As Chieng et al. reported, the addition of plasticizer content led to increase the intensity of tan delta due to higher molecular mobility caused by the plasticizing effect [35]. Then, a significant increase in respect to neat PLA in the magnitude of the damping factor was observed with higher ECO content. On the other hand, Silverajah et al. observed that increasing the content of epoxidized palm oil plasticizer in PLA led to decrease the temperature of tan δ peak [49].

Table 2 shows a summary of values obtained for Vicat softening temperature (VST) and heat deflection temperature (HDT) of neat PLA and PLA plasticized with ECO. These thermomechanical parameters are directly related to mechanical resistant properties. For this reason, the decreasing trend was the same that had been observed previously in tensile strength and modulus. Regarding VST, a remarkable decrease was detected when ECO content increased. Neat PLA had a VST of 56.6 °C, while samples with 7.5 and 10 wt.% ECO, respectively, reduced this value 4.4 °C. A similar trend was observed in HDT values, obtaining a difference of 2.6 °C between neat PLA and PLA_10%ECO sample. The addition of ECO to PLA samples in different percentages facilitated the sliding of the polymeric chains, being also favored by the slight increase in temperature that takes place in the VST and HDT tests. The ECO molecules decreased the intermolecular attraction forces between the PLA and the macromolecules, obtaining the plasticization of the materials [50].

### 3.3. Effect of ECO in PLA on the Thermal Properties of PLA

The addition of ECO led to a change in the main thermal transitions of neat PLA. Neat PLA presented a T_g_ at 62 °C, cold crystallization temperature (T_cc_) at 119.4 °C, and melt temperature T_m_ at 150 °C, as can be seen in Table 3. As percentage of ECO increased in PLA matrix, a remarkable decrease of T_g_ values (62 °C from neat PLA up to 56.8 °C for PLA_10%ECO) was observed. It was seen that PLA_7.5%ECO and PLA_10%ECO showed similar T_g_ values (56.3 and 56.8 °C, respectively), demonstrating the plasticizer saturation effect. Additionally, the T_cc_ values presented a slight decrease with respect to neat PLA, obtaining in all plasticizer formulations lower values than neat PLA. Melt temperature (T_m_) presented also a slight decrease with the incorporation of ECO, concluding that the presence of plasticizer also affected the melt temperature due to the increase of chain mobility. The same evolution was reported by Garcia-Garcia et al., who employed epoxidized Karanja oil in PLA [31]. Addition of ECO allowed to increase the free volume of polymer chains and thus provide better movement at lower temperatures [48]. The changes in the degree of crystallinity (X_c_) also indicated the chain motions of PLA. It was clearly observed that X_c_ of PLA increased with the rise of ECO content, confirming that plasticizer enables the mobility of chains to form stable crystallites at lower energy conditions. Specifically, above 5 wt.% ECO, more evident changes were observed, where the highest crystallinity was found at PLA_10%ECO with 11.5%. This value was almost 67% higher than neat PLA, indicating the enhancement of chain mobility.

Thermogravimetric analysis (TGA) was assessed for PLA formulations with different contents of ECO. Temperature at 5% weight loss (T_5%_) and temperature at maximum degradation (T_max_) showed an important increase in thermal stabilization. CO presented higher thermal stability at elevated temperatures than neat PLA, obtaining in T_5%_ around of 320 °C and T_max_ around of 425 °C, as was reported by Timilsena et al. [33]. On the other hand, in order to study the influence of the epoxidation process in thermal stability of the vegetable oil, the authors compared an epoxidized linseed oil [51] and virgin linseed oil [52], obtaining very similar results. Linseed oil is characterized by an epoxy content very similar to ECO. For this reason, thermal stability of CO could be considered the same as ECO. Thus, as can be seen in Figure 7a, higher contents of ECO provided an improvement of thermal stabilization with respect to neat PLA. Regarding weight loss derivate (Figure 7b), it was observed that samples up to 5 wt.% ECO showed an increase of T_5%_, while higher contents caused a decrease, possibly due to the first evidence of plasticizer saturation [31]. On the other hand, T_max_ of neat PLA was 390.4 °C and increased up to 404.9 °C for PLA formulations with a 2.5–5% ECO, showing a slight decrease when the saturation effect was beginning. Typical plasticizers employed in PLA as citrate esters or ATBC reduce thermal stability of PLA when their content increases [53,54]. However, in this report, different trends were observed in T_5%_ and T_max_. Addition of ECO produced an evident delay of degradation temperature, as was reported by Garcia-Garcia et al., who used epoxidized oils as plasticizers [20]. The reason for this behavior was due to the presence of epoxy groups of ECO that allowed the scavenging of acid groups, obtaining a better thermal stabilization [55].

### 3.4. Disintegration under Composting Conditions

Disintegration process under compost soil of neat and plasticized PLA is shown in both Figure 8 and Figure 9, in which the visual appearance and the weight loss in respect to the initial mass, respectively, are plotted. After 3 days of incubation, film samples changed their visual appearance from translucid to opaque due to increased crystallinity and possible water absorption. It is important to remark that this experiment was conducted at thermophile conditions, with constant temperature of 58 °C and 50% relative humidity. The proximity to the T_g_, as was studied by DSC, can induce an increment of chain mobility and thus the crystallization that causes the increasing opacity [31]. After 7 days buried in controlled compost soil, neat PLA and the sample with less amount of ECO, started the embrittlement process and slight weight loss, as plotted in Figure 9. However, samples with higher amounts of ECO did not show any signs of disintegration. After 14 days buried, visual changes and weight loss were evident in all samples. Neat PLA disintegrates faster than plasticized PLA with ECO. Neat PLA obtained the highest weight loss with 60.2%, while PLA_10%ECO lost 35% of its initial weight. Above 21 days buried in controlled conditions, samples showed a physical inconsistency and disintegration. According to ISO 20200, a disintegrable material was considered when the degree of disintegration achieved 90%. Neat PLA and samples with 2.5 and 5 wt.% of ECO achieved more than 90% of weight loss in respect to their initial value. However, samples with 7.5 and 10 wt. % needed 3 more days to reach this value, indicating a delay of the disintegration process provided by an increasing amount of ECO. These results are in concordance with Balart et al., who observed a reduction of disintegration capacity by increasing the epoxidized linseed oil content in PLA matrix [56]. This delay was directly related with the fact that samples plasticized with ECO possess a higher degree of crystallinity as reported in Table 3. Biodegradability is usually done by lipases, proteases, and esterase secreted from microorganism in the soil compost [57] and this microorganism acts faster in amorphous domains [58,59]. Thus, addition of ECO leads to a delay in the disintegrations process but, in general terms, PLA films developed with ECO can be considered as biodegradable according to ISO 20200.

### 3.5. Migration of ECO by Solvent Extraction Test

PLA is considered a safe polymer for food contact applications [60]. For this reason, the development of a new plasticizer, as ECO, needs to be studied due to plasticizer migration being an important drawback [61,62]. Plasticizer migration is defined as the capacity of transferring plasticizer molecules from the surface of the matrix to the contact medium [63] and the assay developed by solvent extraction is a quite aggressive test to provide information about the potential use at industrial scale. In Figure 10, the percentage of migration of plasticizer from PLA matrix using *n*-hexane as dissolvent at different temperatures is plotted. Neat PLA, taken as control sample, presented a similar value below 0.02% between 30 and 60 °C due to absence of plasticizer. Regarding plasticized PLA with ECO, an increment of percentage of migration is appreciated as temperature increases achieving a highest migration of 0.108% with PLA_7.5%ECO and PLA_10%ECO at 60 °C. These results were lower than those reported by Carbonell-Verdu et al., who employed epoxidized cottonseed oil, obtaining values up to 0.12% [45]. A higher content of epoxy groups in ECO provided stronger interactions with the hydroxyl groups of PLA matrix and, as a consequence, lower migration levels in respect to other plasticizers with less reactive groups, indicating a correct functionality to be employed at an industrial scale. On the other hand, the high molecular weight characterizing the vegetable oils (around 900 g·mol^−1^) could be another positive effect to minimize the migration level, compared with industrial plasticizers with lower molecular weight, as for example tributyl citrate plasticizers (350–400 g·mol^−1^) [48].

## 4. Conclusions

Epoxidized chia seed oil (ECO) was applied in a PLA matrix to evaluate its effect as a new bio-based plasticizer. The low elongation at break of neat PLA (8%) was improved up to values of 64.5% using 10 wt.% ECO, obtaining an improvement of 700%. Morphological images showed spherical voids at equal or higher percentage of 7.5 wt.% ECO in PLA matrix, indicating the beginning of plasticizer saturation. With regard to absorbed impact energy, an almost twice as high value was obtained with 10 wt.% ECO in respect to neat PLA, and, as a consequence, a decrease of hardness. Then, ECO led to enhanced chain mobility of PLA, which induced an increase in free volume and a reduction in intermolecular forces. This lubrication of chains also led to reduce the glass temperature around 4.7 °C with 7.5–10 wt.% ECO and slightly the cold crystallization temperature. Thermal stability was highly improved up to 14.0 °C in respect to neat PLA. The disintegration ability of PLA up to 5 wt.% ECO content was not affected, meanwhile at 7.5–10 wt.%, it was slightly delayed, being considered, in general terms, biodegradable formulations. Finally, very low migration of plasticizer was detected in migration test by using *n*-hexane. The maximum migration recorded was 0.108% with 10 wt.% ECO at 60 °C, while lower migration was obtained (<0.06%) at 30 °C. Therefore, ECO is a promising bio-based plasticizer with potential to be applied in the packaging industry.

## Figures and Tables

**Figure 1 polymers-13-01283-f001:**
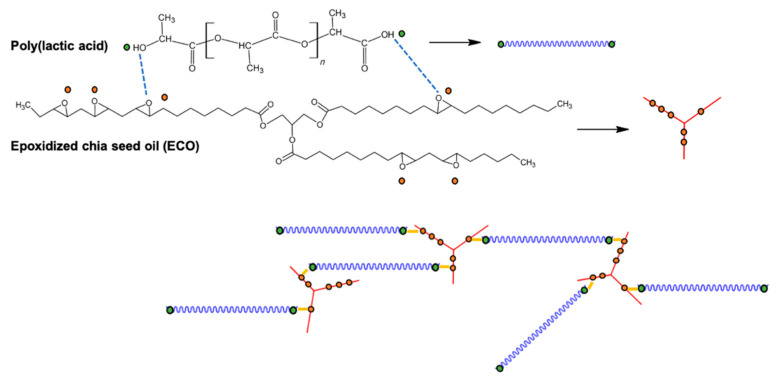
Schematic representation of chemical interactions between PLA and ECO.

**Figure 2 polymers-13-01283-f002:**
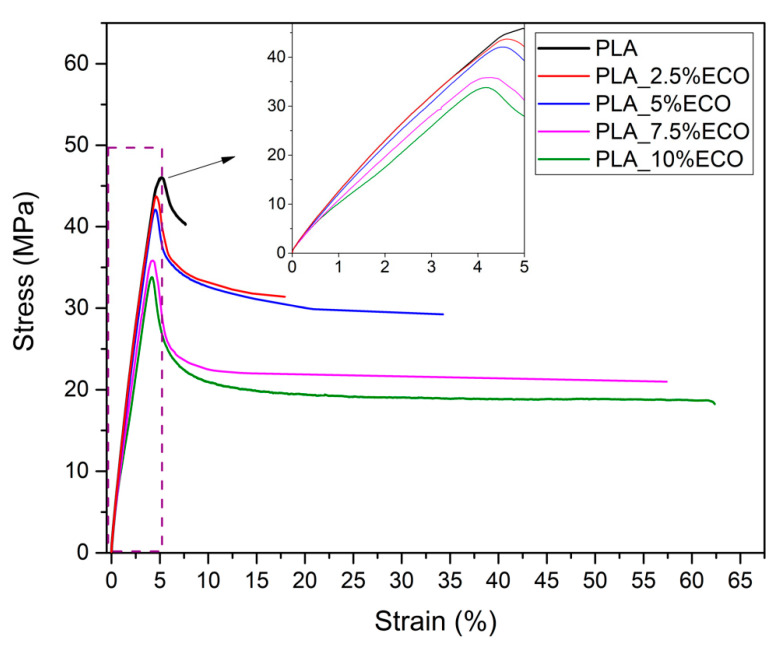
Plot evolution of characteristic stress-strain curves of PLA with different epoxidized chia seed oil (ECO) contents.

**Figure 3 polymers-13-01283-f003:**
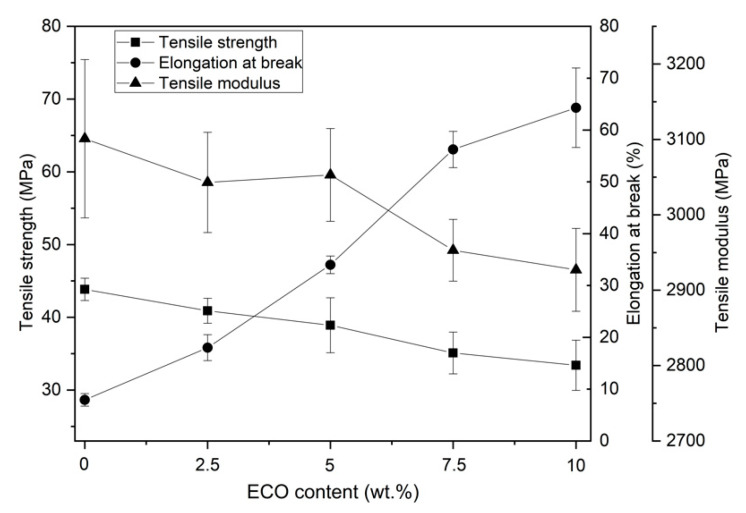
Plot evolution of tensile mechanical properties of PLA with different epoxidized chia seed oil (ECO) contents.

**Figure 4 polymers-13-01283-f004:**
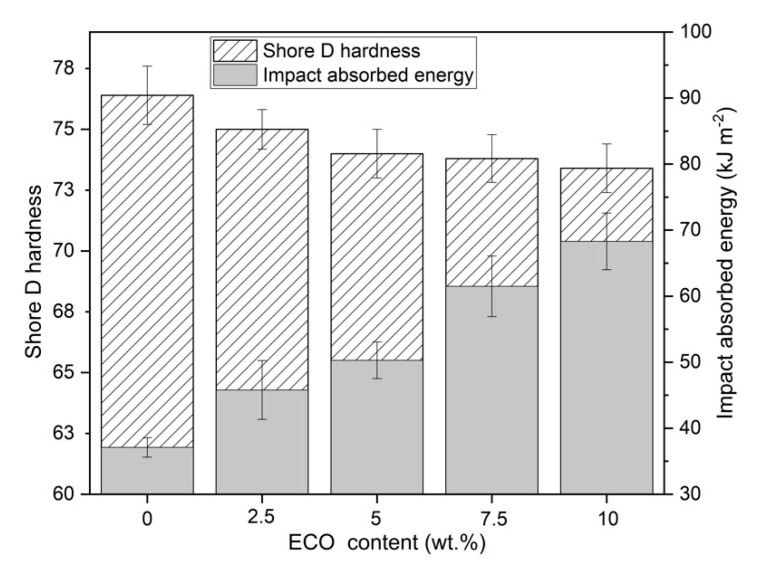
Plot evolution of Shore D hardness and impact absorbed energy of PLA with different epoxidized chia seed oil (ECO) content.

**Figure 5 polymers-13-01283-f005:**
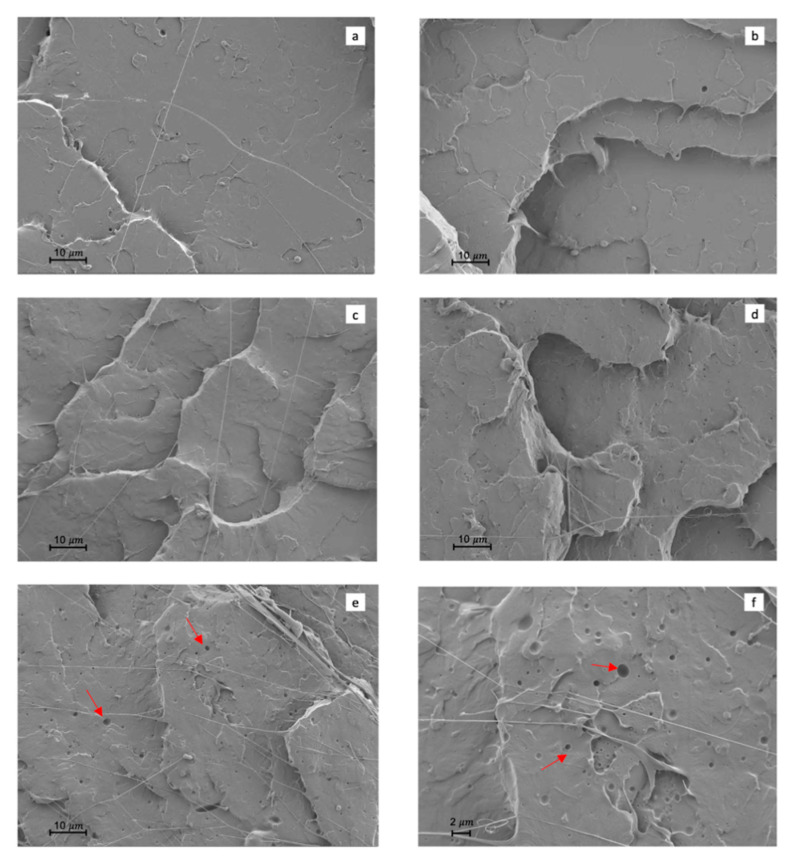
Fracture surface morphology of Charpy test at 1000× by field emission scanning electron microscopy (FESEM): (**a**) neat PLA; (**b**) PLA_2.5%ECO; (**c**) PLA_5%ECO; (**d**) PLA_7.5%ECO; (**e**) PLA_10%ECO; and (**f**) PLA_10%ECO at 2500×.

**Figure 6 polymers-13-01283-f006:**
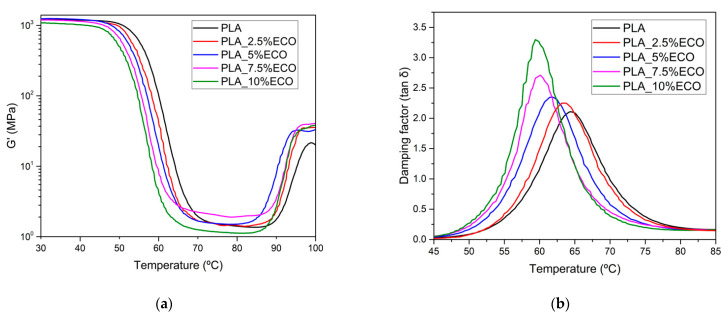
Plot evolution of dynamic mechanical thermal analysis (DMTA) of PLA with different ECO contents: (**a**) storage modulus (G’); (**b**) damping factor (tan δ).

**Figure 7 polymers-13-01283-f007:**
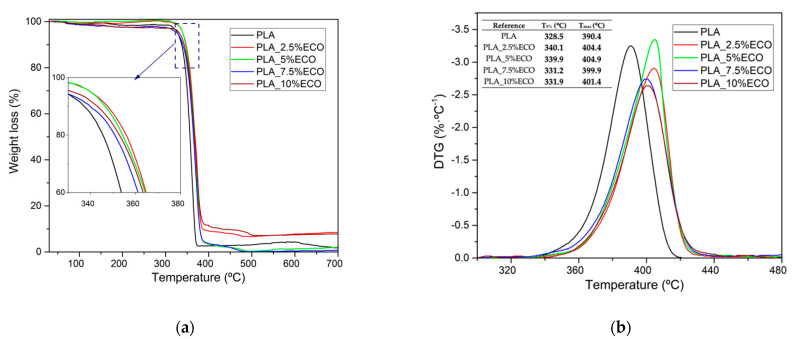
Thermal parameters of degradation of PLA with different epoxidized chia seed oil (ECO) contents. (**a**) Weight loss; (**b**) derivative thermogravimetry. T_5%_ is temperature at 5% weight loss; T_max_ is temperature at maximum degradation.

**Figure 8 polymers-13-01283-f008:**
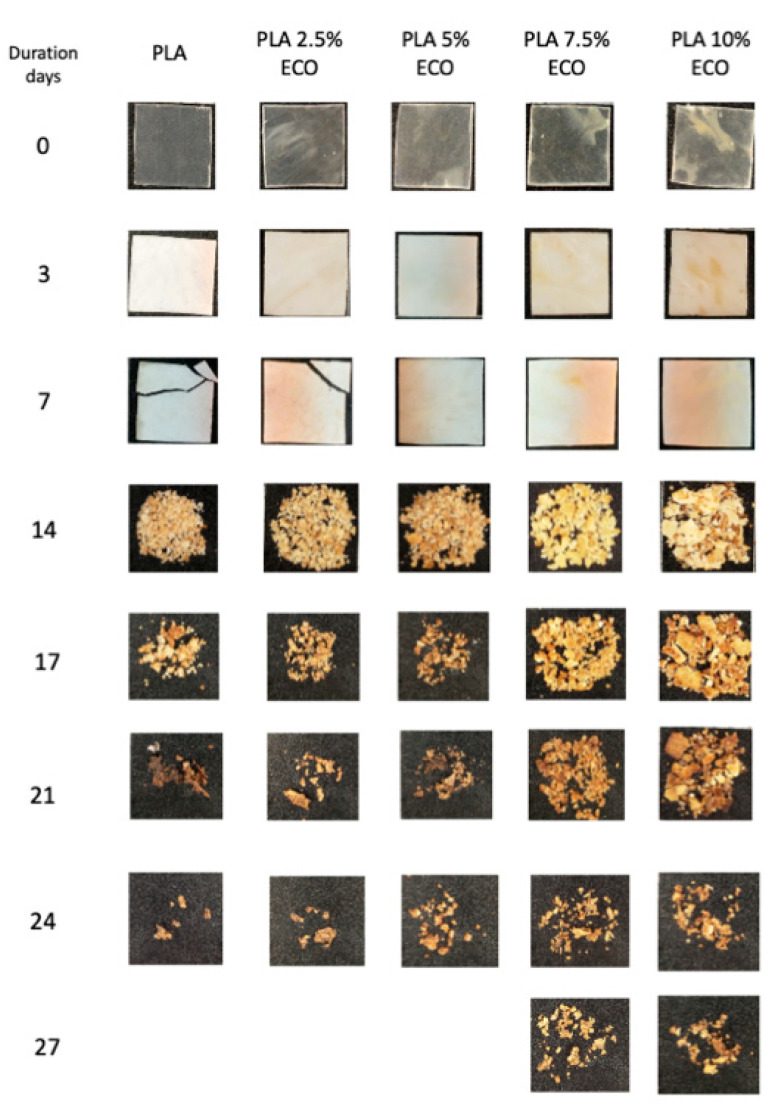
Visual appearance of disintegration under composting conditions.

**Figure 9 polymers-13-01283-f009:**
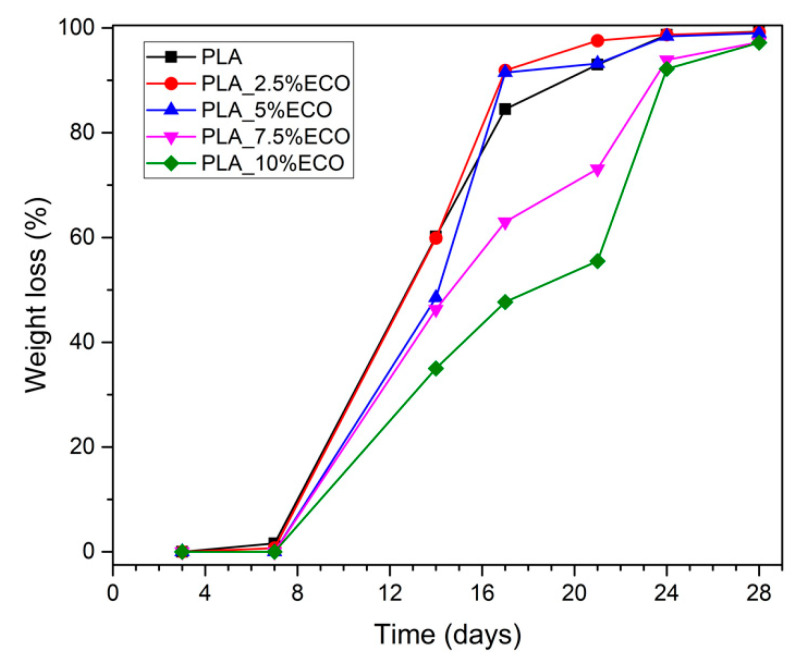
Weight loss recorded during disintegration test.

**Figure 10 polymers-13-01283-f010:**
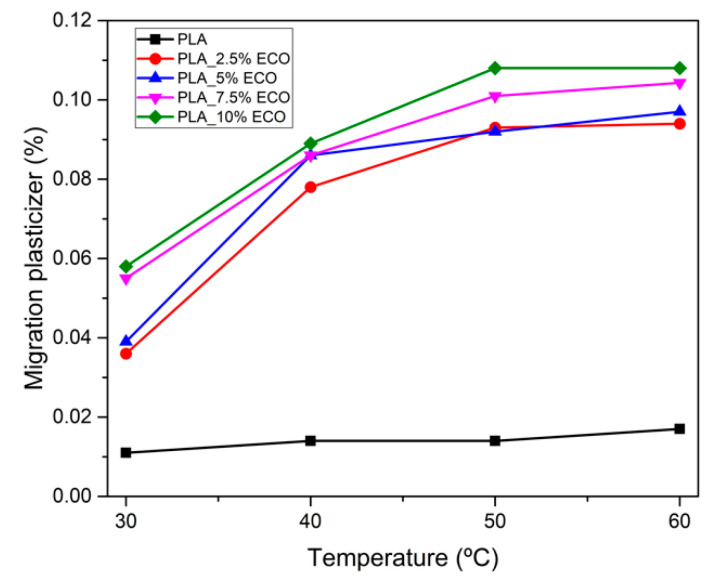
Migration of epoxidized chia seed oil (ECO) plasticizer in PLA matrix by n-hexane solvent extraction.

**Table 1 polymers-13-01283-t001:** Composition of ECO plasticized PLA materials and labelling.

Reference	Parts by Weight (wt.%)	
	PLA	ECO
PLA	100	0
PLA_2.5%ECO	97.5	2.5
PLA_5%ECO	95	5
PLA_7.5%ECO	92.5	7.5
PLA_10%ECO	90	10

**Table 2 polymers-13-01283-t002:** Vicat softening temperature (VST) and heat deflection temperature (HDT) of PLA with different epoxidized chia seed oil (ECO) content.

Reference	VST (°C)	HDT (°C)
PLA	56.6 ± 1.5	52.8 ± 0.5
PLA_2.5%ECO	55 ± 1.2	51.8 ± 0.6
PLA_5%ECO	53.2 ± 1.3	50.4 ± 0.4
PLA_7.5%ECO	52.2 ± 1.1	50.4 ± 0.6
PLA_10%ECO	52.2 ± 1.4	50.2 ± 0.4

**Table 3 polymers-13-01283-t003:** Main thermal parameters of PLA plasticized with different ECO contents obtained using DSC.

Reference	T_g_ (°C) ^1^	T_cc_ (°C) ^2^	$∆H_{cc}\;\left(J\cdot g^{-1}\right){\;}^{3}$	T_m_ (°C) ^4^	∆*H_m_* (J·g^−1^) ^5^	$X_{c}\;\left(\%\right){\;}^{6}$
PLA	62.0	119.4	8.00	150.0	15.0	7.5
PLA_2.5%ECO	60.0	117.3	12.5	149.7	19.6	7.9
PLA_5%ECO	59.2	117.7	10.9	149.0	18.4	8.5
PLA_7.5%ECO	56.3	118.0	11.6	148.2	20.3	10.1
PLA_10%ECO	56.8	118.5	8.4	148.4	18.0	11.5

^1^ Glass transition temperature; ^2^ Cold crystallization temperature; ^3^ Cold crystallization enthalpy; ^4^ Melt temperature; ^5^ Melt enthalpy; ^6^ Degree of crystallization.

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
