# Peer review of "Dual Plasticizer/Thermal Stabilizer Effect of Epoxidized Chia Seed Oil (*Salvia hispanica* L.) to Improve Ductility and Thermal Properties of Poly(Lactic Acid)"

_polymers, 2021, doi:10.3390/polym13081283_

Round 1
Reviewer 1 Report
"To measure the weight loss due to the disintegration process, samples were periodically unburied (3, 7, 14, 17, 21, 24 and 28 days), washed with distilled water and dried 24 h before to be weighed in an analytical balance." - Did Authors took out the samples, washed them, dried, weighted and buried again? Or samples after unburying were tossed out? Because it seems strange to take out the sample, clean it, stop degradation and put it in again claiming that it is ongoing process.
Was control sample subjected to hexane extraction test for comparison?
Discussion of tensile test result could include the discussion about the materials' toughness? Was it improved by the ECO addition?
Figure 5 - please present the linear plots, dotted ones are very similar, especially that legend is hardly visible.
Please provide the discussion about the changes in tan delta peak magnitude.
Table presented as a part of Figure 6 is hardly visible.
What is the reason of such irregularities in T5%? Why it was decreasing for higher contents of ECO?
Please present the thermal stability of applied plasticizer.
Author Response
In the attached document, reviewer can find alls changes made following their considerations.

Reviewer 2 Report
The work is well planned and well presented. Effects of epoxidized chia seed oil (ECO) as a plasticizer in PLA can be clearly seen from the results of authors' work.
Author Response
The authors would like to thank Reviewer 2 for his/her comments.

Reviewer 3 Report
This work investigated the effect of epoxidized Chia seed oil on the mechanical and thermal properties of PLA. The manuscript can be accepted after the major revision.
- How about the content of hydroxyl group and carboxyl group for the used PLA?
- Did the epoxy group of EOC react with the hydroxyl group or carboxyl group of PLA? The evidence should be given.
- How did the author determine the epoxy content of EOC? Can the ECO with different contents of epoxy be prepared?
- The typical stress-strain curves of PLA with different ECO contents should be given.
Author Response
In the attached word document, reviewer can find all changes made following his/her considerations.

Round 2
Reviewer 1 Report
Everything in order after revision